# A Scalable Self-supervised Learner for Hyperspectral Image Classification

**Weili Kong, Lin Qi & Baisen Liu**[*]
Information and Communication Engineering
Harbin Engineering University
{kkweil, qilin, liubaisen}@hrbeu.edu.cn

**Jiaming Pei**
School of computer science
The University of Sydney
jpei0906@uni.sydney.edu.au

## Abstract

Learning-based Hyperspectral image classification methods have achieved fantastic performance due to their superior ability to represent features. At the cost, these methods are complex, inflexible and weak to generalize. Thus, we proposed a simple and scalable pretrained model which can extremely accelerate the convergence rate and promote the performance in downstream task. The results of the experiment indicated that our method has achieved state-of-the-art performance with limited training samples in classification task.

## 1 Introduction

Hyperspectral image(HSI) is a kind of image which greatly differs from RGB image, it has hundreds of channels and contains rich spectral information. Recently, some studies combined CNN and transformer models which have achieved excellent performance in HSI classification tasks(He et al. (2019),Sun et al. (2022)). The structure and output of convolutional networks are controlled by the input, consequently, the generation ability of the aforementioned methods is limited. Inspired by self-supervised model such as BEiT(Bao et al. (2021)) and MAE(He et al. (2022)). We designed a transformer based self-supervised leaner for HSI, which can generalize to arbitrary input size and spectral resolution and accelerate the downstream training process. Moreover, we employed an auxiliary task to incorporate domain prior knowledge into the model, as shown in Figure 1.

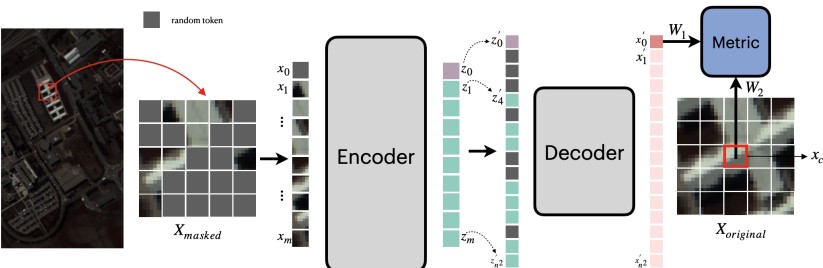

Figure 1: The structure of self-supervised leaner (Masked on pixel-wise)

## 2 Methodology

For the analytic of hyperspectral images, humans are primarily concerned with the central pixel of the patch, while the transformer-based model performs feature extraction on the patch globally. Therefore, we teach the model, through an auxiliary task, how to infer the information embedded in the central pixel from the global information of the patches, thus model naturally learns this domain prior. This helps to overcome the defect of transformer-based model which lack inductive bias(Vaswani et al. (2017)). Specifically, we first random mask and flatten the original patch $X \in$

---

[*]Corresponding author

$\mathbb{R}^{n \times n \times c}$, then contact a random token $\boldsymbol{x}_0$ to it as the encoder input. After that, we project $\boldsymbol{x}_c$ and $\boldsymbol{x}_0'$ into some metric space and minimize the distance between the projected vectors. The loss function of the pretraining section is

$$l = \frac{1}{n^2} \sum_{i=1}^{n^2} ||\boldsymbol{x}_i - \boldsymbol{x}_i'||^2 + ||\boldsymbol{W}_1 \boldsymbol{x}_0' - \boldsymbol{W}_2 \boldsymbol{x}_c||^2 \tag{1}$$

where $\boldsymbol{x}_c$ represents the center pixel, $\boldsymbol{x}_0'$ represents the decoder output of random token $\boldsymbol{x}_0$, $\boldsymbol{x}_i$ and $\boldsymbol{x}_i'$ represent each pixel and its reconstruction, $\boldsymbol{W}_1$ and $\boldsymbol{W}_2$ represent the project matrix.

To put the tokens into fullplay in downstream task, instead of global average pooling, we utilize a learnable aggregation. Let the output of encoder $\mathbb{Z} = \{\boldsymbol{z}_0, \boldsymbol{z}_1, \boldsymbol{z}_2, \ldots, \boldsymbol{z}_m \,|\, \boldsymbol{z} \in \mathbb{R}^d\}$, the final logit $\boldsymbol{v}$ can compute as follow equations:

$$\boldsymbol{M} = [\boldsymbol{z}_1, \boldsymbol{z}_2, \ldots, \boldsymbol{z}_m]^T, \boldsymbol{M} \in \mathbb{R}^{m \times d} \tag{2}$$

$$\boldsymbol{M}' = [f(\boldsymbol{z}_1), f(\boldsymbol{z}_2), \ldots, f(\boldsymbol{z}_m)]^T \tag{3}$$

$$\boldsymbol{v} = \boldsymbol{M}^T \boldsymbol{M}' g(\boldsymbol{z}_0) + \boldsymbol{z}_0, \boldsymbol{v} \in \mathbb{R}^d \tag{4}$$

where $f$ and $g$ represent MLP mapping, $d$ represents the embedding dimension of encoder.

## 3 EXPERIMENTAL

**Dataset.** We divided HSIs which captured by GF-5 satellite into patches of 4 different sizes as the pretraining data, the total pretraining samples is about 300 thousand. We fine-tuned the pretrained model on three widely used datasets which are Salinas(SA), Indian Pines(IP) and PaviaU(PU), and the train-test ratio was set to 1:9. The detail of datasets is attached in Appendix A.1.

**Hyper-parameters.** We used MAE as the baseline, the mask ratio, embedding dimension, depth, heads of encoder were set to 0.5, 256, 4, 8 respectively, the decoder was half of it. We used AdamW as optimizer, the learning rate was set to $8 \times 10^{-4}$ in pretrain stage. In downstream task, the learning rate of encoder and classifier were set to $10^{-5}$ and $10^{-3}$ respectively.

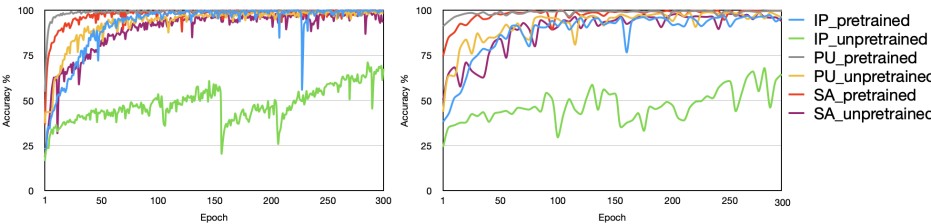

Figure 2: Accuracy curve in training process(Left: Training curve, Right: Testing curve)

**Quantitative Evaluation.** In classification task, as the Figure 2 shown, our method can greatly improve the performance and speed up the convergence rate especially when the training data is relatively small. The classification performance and comparison with other start-of-the-art methods are attached in Appendix A.2.

**Influence of auxiliary task.** The auxiliary task can be considered as a regularization constraint on the pretrained model, forcing it to extract features in the global information of patch that are significantly linked with the central pixel. In this way, the scope of the model's functional domain is narrowed, it is possible to significantly minimize the quantity of training data, which enhances the model's performance in the case of data scarcity.

**Conclusion.** In this work, we designed a pretrained model for HSI classification which can break the limitation of input size and channel. Furthermore, we constructed an auxiliary task to overcome the large training data requirements when training transformer model. Our methodology exhibited strong generalization ability and achieved outstanding performance on downstream tasks. In future work, we plan to use this idea for few-shot classification as well as unsupervised clustering.

ACKNOWLEDGEMENTS

The work was supported by the Natural Science Foundation of Heilongjiang Province for Key projects, China, and the Postdoctoral Scientific Research Developmental Fund of Heilongjiang Province, China.

URM STATEMENT

The authors acknowledge that at least one key author of this work meets the URM criteria of ICLR 2023 Tiny Papers Track.

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

## A APPENDIX

### A.1 DATASET

**Pretraining Dataset.** To train the model, we selected hyperspectral images from a variety of scenes, including forest, city, desert, mountain village, rural, Gobi, and snow, and divided them into four patches of varying size, 9, 15, 29, 33 respectively. These HSIs were gathered by GF-5 satellite which contain 297 spectral bands in the wavelength range $0.4$–$2.5 \times 10^{-6}$m. The spatial resolution of these data is 30m.

**Indian Pines.** The Indian Pines data set contains 145×145 pixels which gathered by the AVIRIS sensor in Northwestern Indiana, where AVRIS stands for airborne visible infrared imaging spectrometer. The original Indian Pines data set contains 220 spectral channels in the wavelength range from $0.4$–$2.5 \times 10^{-6}$m with a spatial resolution of 20m. In this paper, 20 bands corrupted by water absorption effects are discarded.It contains 16 classes and 42776 labeled pixels in total.

**Salinas.** The Salinas data set contains 512×217 pixels also collected by the AVIRIS sensor over Salinas Valley, California. These data contain 224 spectral bands range from $0.4$–$2.5 \times 10^{-6}$m with a spatial resolution of 3.7m. It contains 16 classes and 50929 labeled pixels in total.

**PaviaU.** The University of Pavia data set contains $610 \times 340$ pixels collected by the ROSIS sensor at the University of Pavia, where ROSIS stands for reflective optics system imaging spectrometer. This image scene contains 103 spectral bands in the wavelength range from $0.43$–$0.86 \times 10^{-6}$m with a spatial resolution of 1.3 m. The data set was provided by Prof. Paolo Gamba from the Telecommunications and Remote Sensing Laboratory, University of Pavia. It contains 9 classes and 42776 labeled pixels in total.

## A.2 EXPERIMENT RESULTS AND COMPARISON

In order to quantify the classification performance of our method, the overall accuracy (OA), average accuracy (AA) and kappa coefficient (Kappa) were employed as evaluation measures. OA is the ratio of the number of correctly labeled hyperspectral pixels to the total number of hyperspectral pixels in test samples. AA is the mean of accuracy in different land-cover categories. Kappa measures the consistency between classification results and ground truth. The larger values of OA, AA, and Kappa represent the better classification results.

To verify the effectiveness of the our method, We compared the classification results with SSAN(Sun et al., 2019), 3DSA-MFN(Qing et al., 2022), SST-FA(He et al., 2021).The size of the input HSI patch for all methods was set to 15×15.

Table 1: Comparison of other methods

| Method | Salinas | | | Indian Pains | | | PaviaU | | |
|---|---|---|---|---|---|---|---|---|---|
| | OA | AA | Kappa | OA | AA | Kappa | OA | AA | Kappa |
| SST-FA | 94.94 | 93.05 | 94.32 | 88.98 | 68.15 | 86.7 | 93.37 | 85.01 | 97.49 |
| SSAN | 96.81 | 98.33 | 96.54 | 95.49 | 94.17 | 94.85 | 98.02 | 96.90 | 97.37 |
| 3DSA-MFN | 99.72 | 99.32 | 99.13 | 96.02 | 93.89 | 94.78 | 98.96 | 96.32 | 97.49 |
| Ours | 99.85 | 99.73 | 99.75 | 96.55 | 93.12 | 96.10 | 99.73 | 99.69 | 99.46 |

We sampled patches of three different sizes from Salinas, Indian Pains and PaviaU which vary from the pretraining dataset to examine the generation capacity of our pretrained model. Train-test split strategy is same as section 3. The classification results are shown in Figure 3. It can be seen that despite the fact that the size and spectral resolution of the training data in the downstream tasks are not consistent with those in the pretraining dataset, our method still achieves excellent classification results on these data.

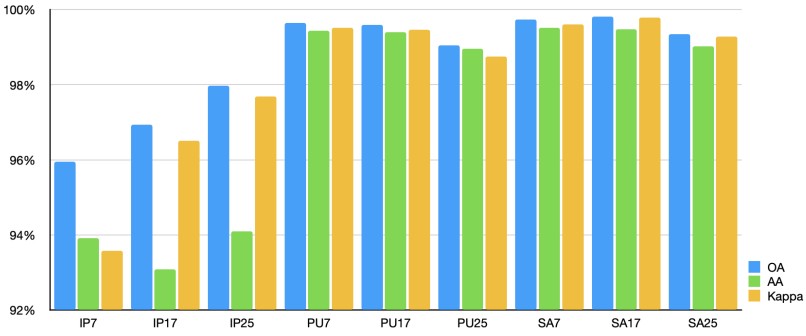

Figure 3: Accuracy of training data with different patch size.

