# OpenReview forum: "A Scalable Self-supervised Learner for Hyperspectral Image Classification"
_ICLR.cc/2023/TinyPapers — Submitted to Tiny Papers @ ICLR 2023_

### Official Review · Reviewer_2FwC · 2023-03-26

**Confidence:** 5

**Summary Of Contributions:**

A pretraining and downstream finetuning technique for hyperspectral image classification is proposed in this paper

**Rating:**

Needs Clarification (NC): a submission which does not meet the reviewing criteria and needs clarification for its described problem or solution

**Strengths And Weaknesses:**

S1: The paper clearly shows the hyperparameters used to train the model.

S2: It can be seen from the experiments that the proposed method can improve performance.

W1: It is not very clear to me why the proposed auxiliary task works for the consider scenario, hyperspectral image classification. The paper should be clear to its readers of the motivation of the method design

W2: The paper is not properly anonymized, the acknowledge section contains information that can be used to identify the author.


**Suggested Changes:**

C1: I would suggest the author to add more about the motivation of the designed method, right now it does not give any explanations of why such a model would work well for the given task.

C2: I would also suggest adding comparisons to BeiT and MAE pretraining.

C3: In the introduction it is claimed that the proposed method can generalize to arbitrary size and spectral resolution, yet the paper does not have the experiments to valid that. I would strongly suggest to at least valid all the claims made in the paper.

---

> ### Author Response · Authors · 2023-05-17
> **Response to Reviewer 2FwC**
>
> **I would like to extend my sincere thanks to the reviewer 2FwC for his time and effort in reviewing my work and providing valuable feedback.**
>
>
> We have made the revisions listed below in response to your thoughtful suggestions:
> 1. Our **motivation** of this work is that design a simple and scalable pertrained model for Hyperspectral image(HSI) classification and can transfer to other HSIs which with different data structures with the source data. Limited by CNN, however, this is hard to achieve.
> + The **motivation**  of our work is mentioned in *Abstract* as follows:
> > At the cost, these methods are complex, inflexible and weak to generalize. Thus, we proposed a simple and scalable pretrained model which can extremely accelerate the convergence rate and promote the performance in downstream task.
>
> + The **motivation**  of our work is mentioned in *Introduction* section as follows:
> > The structure and output of convolutional networks are controlled by the input, consequently, the generation ability of the aforementioned methods is limited. Inspired by self-supervised model such as BEiT(Bao et al. (2021)) and MAE(He et al. (2022)). We designed a transformer based self-supervised leaner for HSI, which can generalize to arbitrary input size and spectral resolution and accelerate the downstream training process.
>
> + In *Methodology* section, we expanded on the rationale for **why it works effectively**. As follows:
> >For the analytic of hyperspectral images, humans are primarily concerned with the central pixel of the patch, while the transformer-based model performs feature extraction on the patch globally. Therefore, we teach the model, through an auxiliary task, how to infer the information embedded in the central pixel from the global information of the patches, thus model naturally learns this domain prior. This helps to overcome the defect of transformer-based model which lack inductive bias.
>
> 2. The main difference between MAE and BEit is that MAE reconstructs at the image level while BeiT reconstructs at the feature level. They are both essentially VIT-based visual pre-training models and they performed similarly on downstream tasks. Given BeiT's more complex structure and our limited computing resources training a model is very time consuming, we feel that it is not worthwhile to train a network using BeiT.
>
> 3. This is our negligence in not detailing the differences between the pre-training data  and the downstream task data in the paper. In fact, they inherently have different spectral and spatial resolutions. We elaborate on the information of each dataset in the *Appendix A.1*.  For examine the generalization capacity to the arbitrary input size, we sampled patches of three different sizes from downstream datasets which vary from the patch sizes in pretraining dataset. The exam results can be seen in Figure 3.
>
>
> **Once again we sincerely appreciate the reviewer's efforts in offering such thorough and considerate criticism on my paper. These thoughtful comments open the door for our higher-quality article. We have carefully revised the questions in the article based on your valuable opinions. If you have any more questions or concerns, please don't hesitate to contact us. Please read the paper's newly amended version, which has been uploaded.**

---

### Official Review · Reviewer_TZx4 · 2023-03-29

**Confidence:** 4

**Summary Of Contributions:**

In this paper, the authors focus on the problem of learning-based hyperspectral image classification.  To this end, this paper proposes a self-supervised learner which can generalize to input patches of different dimensions. A scalable pretrained model is presented to improve performance on downstream classification tasks. This model has been fine-tuned on three different datasets and achieves state-of-the-art performance.

**Rating:**

Great Start (GS): a submission which meets some of the reviewing criteria but has room for improvement

**Strengths And Weaknesses:**

**Strengths**
* This paper proposes a method to pre-train models on HSI data, using an encoder-based self-supervised learner.
* The method consists of augmenting a flattened image patch with a random token, using it during the pretraining section. The representation learned by the encoder is combined in a learnable combination using MLPs. Thus, the network can take advantage of domain knowledge.
* The experiments cover fine-tuning on three datasets. This method achieves SOTA performance by using only 10% of the data for training.
* The method has been compared to three SOTA methods by using different metrics.
* Lastly, the paper is written in a clear and concise manner. The finding is communicated effectively.

**Weaknesses**
* Some hyperparameters such as the patch size used for pre-training can be added in the hyperparameters section. I found these additional details lacking.
*  I find that the details provided in the paper are not sufficient to reproduce it. Without open-source code, since some details are missing, the results cannot be reproduced by others working on the same problem.
* Some grammatical errors are typos need to be fixed (mentioned in *Suggested Changes*)
* The grant information has not been removed

**Summary of the Review**

This paper aims to introduce a pre-training method for hyperspectral image classification which has the ability to generalize. Overall, the paper communicates the findings clearly and effectively. The claims appear to be correct to the best of my knowledge. The paper does not include enough information to be reproduced. Lastly, it follows the basic requirements for ICLR submission.


**Suggested Changes:**

Overall, the paper is well-written. The following changes should be made:
* Given the limited space for this format, Figure 1 can be made more descriptive by adding the names of vectors and latent variables. It would help the readers in understanding the equations.
* There are some typos which need to be fixed. For example, in the “Experimental” section, the word “we use” appears twice in the last sentence of the first paragraph. In the “Experimental” section, the last line of the Quantitative Evaluation section has the typo “SOAT methods.”

---

> ### Author Response · Authors · 2023-05-17
> **Kindly response to reviewer TZx4**
>
> **I am grateful to the reviewer TZx4 for his insightful comments and suggestions, which have greatly improved the quality of my work.**
>
> Following your precious suggestions, we have mede the changes as below:
>
> 1. We add a subsection A.1 where a detailed description of the pre-training dataset we used in Appendix section.
> 2. Based on your suggestion, we have added some annotations in Figure 1 to help readers understand.
> 3. We carefully checked the grammatical problems of the full text and made corrections, and the errors in your suggestion have been corrected.
> 4. To allay worries about non-anonymity, the acknowledgments section has been removed.
> 5. We are sorting out the code and will publish it on [*github*](https://github.com/kkweil/hsi-pretrian-cls.git) in a few days.
>
>
> **We sincerely appreciate the reviewer's efforts in offering such thorough and considerate criticism of our manuscript. These insightful comments pave the way for articles of higher caliber. Now that they have all been fixed. Do not hesitate to contact us if you have any further queries or worries. Please review the newly revised version of the paper that has been uploaded.**

---

### Official Review · Reviewer_VZzV · 2023-03-30

**Confidence:** 3

**Summary Of Contributions:**

The paper has proposed a flexible, simple and scalable method for classifying hyperspectral images. The authors have proposed the idea of adding an auxiliary task to counter the lack of inductive bias in SOTA transformer models.

**Rating:**

Clear, Correct, and Reproducible (CCR): a submission which meets the reviewing criteria

**Strengths And Weaknesses:**

Strengths :

1. The paper is well-written, very well-structured and replete with necessary experimental results.
2.  It focuses on an important topic of classifying hyperspectral images by augmenting available domain knowledge to improve performance.
3. The concept put forward in the paper is very well-represented with proper explanations.
4.  The results claimed by the authors show improvement over state-of-the-art methods.

Weaknesses :

1. I think the paper is more or less flawless barring one or two spelling mistakes.
2. The acknowledgements section of the paper contains information which might breach the anonymity of the paper.




**Suggested Changes:**

I thought that the paper is well-written except a few typos.The acknowledgements section should be removed. Also, a short concluding remark with a scope for future work would be handy.

---

> ### Author Response · Authors · 2023-05-17
> **Response to Reviewr VZzV**
>
> **I would like to express my heartfelt gratitude to the reviewer VZzV for their invaluable feedback and constructive comments.**
>
> According to the valuable suggestions of reviewer VZzV, we have made the following revisions:
> 1. We carefully checked the articles for grammatical and spelling errors and corrected them.  In addition, we have polished the presentation of the language.
> 2. We have removed the acknowledgments section to eliminate the non-anonymity concerns.
> 3. We have added a brief summary at the end of the paper, as follows
> >In this work, we designed a pretrained model for HSI classification which can break the limitation of input size and channel. Furthermore, we constructed an auxiliary task to overcome the large training data requirements when training transformer model. Our methodology exhibited strong generalization ability and achieved outstanding performance on downstream tasks. In future work, we plan to use this idea for few-shot classification as well as unsupervised clustering.
>
> **We are deeply appreciative of the reviewer's efforts in providing such detailed and thoughtful feedback on our manuscript. These valuable comments lead a path to higher quality articles.  All of them have been corrected now.  If you have any additional questions or concerns, please do not hesitate to let us know. The new revised version of the paper has uploaded, please check it.**

---

### Meta-Review · Area_Chair_bHdv · 2023-04-06

**Recommendation:** Invite to archive
**Confidence:** 5

**Metareview:**


This paper proposes a self-supervised learning method for the task of hyperspectral image classification.
The method is designed to be able to incorporate domain knowledges of the hyperspectral image by using MLPs and an auxiliary task.

One reviewer thinks this paper is CCR, yet the other two reviewers all raise concerns about the clarity and the reproduciabilty of the paper.
It is suggested that the paper need to take into consideration of the concerns of the reviewers and revise the paper.


**Summary:**

A method for pretraining model for hyperspectral image classification is proposed, it is shown in the paper that the proposed method can achieves SOTA performance using less supervised data, yet the main concerns from the reviewers are not negligible, it is recommended to revise the paper for archive.

**Reason For Not Giving A Higher Recommendation:**

Two out of three reviewers think the paper are not CCR, some details are needed for revising.


**Reason For Not Giving A Lower Recommendation:**

This paper can achieve SOTA performance, and the method are considered to be effective by all reviewers.

---

### Decision · Program_Chairs · 2023-04-10

Invite to archive